# Assessing Population Well-Being in Saudi Arabia: A Comprehensive National Survey Using the WHO-05 Index and Self-Rated Health Metrics

**DOI:** 10.3390/healthcare13030310

**Published:** 2025-02-03

**Authors:** Abdulaziz Alahmadi, Yasir Almuzaini, Majed Alosaimi, Ahmed Alahmari, Fahad Alamri, Haytham Sheerah, Mariyyah Alburayh, Anas Khan

**Affiliations:** 1Department of Preventive Medicine, Ministry of National Guard-Health Affairs, Riyadh 11426, Saudi Arabia; 2Department of Public Health, College of Public Health, Imam Abdulrahman Bin Faisal University, Dammam 34212, Saudi Arabia; 3Saudi Public Health Authority, Riyadh 13351, Saudi Arabia; 4Global Centre for Mass Gatherings Medicine, Ministry of Health, Riyadh 12372, Saudi Arabia; 5Office of the Vice Minister of Health, Ministry of Health, Riyadh 11451, Saudi Arabia; hasheerah@moh.gov.sa; 6Department of Emergency Medicine, College of Medicine, King Saud University, Riyadh 11451, Saudi Arabia

**Keywords:** WHO-5 index, self-rated health status, well-being, service quality and performance, social determinants of health (SDH)

## Abstract

Introduction: The well-being of populations is crucial for understanding nations’ public health and progress. With its ongoing transformation and development objectives, Saudi Arabia emphasizes its residents’ quality of life and well-being. Recent surveys and health indicators have underlined the nation’s focus on enhancing population well-being. Aim: This study aimed to assess the overall level of well-being of the population living in Saudi Arabia using the WHO-5 index and self-rated health (SRH) metrics. Subject and methods: This cross-sectional study was conducted among thirty-nine thousand five hundred and sixty-two people from the general population in Saudi Arabia, citizens and residents, including all adult age groups (>18 years) and both genders. A self-administered questionnaire was sent to the Saudi adult population using an online survey. The questionnaire includes socio-demographic characteristics, the five-item Well-Being Index (WHO-5), and the self-rated health status. Results: Based on SRH, 77.4% were considered a healthy group. Male gender participants who had a better education were more likely to report a favorable SRH. According to WHO-5, poor well-being was seen in 26% of the population, and the rest had good well-being (74%). Independent predictors for good well-being include increasing age and educational level and being unemployed. Interestingly, we found a positive significant correlation between SRH and WHO-5 scores (r = 0.371; *p* < 0.001). Conclusions: Poor well-being was common among the general population. Independent risk factors for poor well-being include female gender and Saudi nationality, while increasing age, male gender, and higher education were significant predictors of healthy SRH. More longitudinal studies are needed to extract more data on this nation’s growing prevalence of poor self-rated health status.

## 1. Introduction

The well-being of populations is crucial for understanding nations’ public health and progress. With its ongoing transformation and development objectives, Saudi Arabia emphasizes its residents’ quality of life and well-being. Recent surveys and health indicators have underlined the nation’s focus on enhancing population well-being [1]. As the global understanding of well-being shifts from an absence of diseases to a holistic understanding of mental, physical, and social health, the WHO-05 index and self-rated health (SRH) have become internationally recognized tools for the assessment of well-being. The World Health Organization’s Five-item Well-Being Index (WHO-5) is a widely recognized and utilized short self-reported screening tool designed to assess population well-being, public health, and overall quality of life. The WHO-5 was first published in 1998; since then, it has been translated into 30 languages and used by researchers worldwide [2]. The validated questionnaire comprises five simple questions within a timeframe of the previous two weeks that assess the well-being of individuals. The inquiries cover areas such as interest, engagement, and mood [3,4]. In various studies, the WHO-5 score has shown a correlation with other psychological parameters, including depression, anxiety, stress, and overall mental health [5].

Self-reported health is one of the most commonly used measures of perceived health status that encompasses a person’s biological, mental, social, and functional aspects. It has been found to be a reliable indicator of overall health and a predictor of mortality. It is also one of the health monitoring indicators recommended by the European Union Commission and the WHO [6]. This measure provides participants with an opportunity to evaluate their own health from a personal standpoint. Participants typically respond to a straightforward question, such as “In general, how would you rate your health?” using a 5-point scale with options ranging from “excellent” to “poor” [7]. Given its simplicity, SRH has achieved significant attention in several research fields due to its reliability and strong predictive validity in predicting key health outcomes, including mortality and the onset of chronic diseases [7].

## 2. Literature Review

Several studies documented associations between well-being and SRH, along with other psychological disorders. For instance, Wuorela et al. (2020) found significant correlations between SRH and objective health status (OH), and OH was considered a strong predictor of mortality during 27-year follow-up than SRH [7]. A large cohort study conducted among 18,000 residents in China reported that life and work pressure, poor spiritual status, and poor quality of interpersonal relationships were associated with poorer SRH [8]. In Canada, comparing participants without anxiety and depression, participants with mild to moderate/severe anxiety and depression were associated with an increased likelihood of having poor SRH [9]. On the contrary, a cross-sectional study conducted in Romania predicted that higher well-being was associated with lower depression and loneliness rates [10]. Supporting these reports, a large cross-sectional study conducted in Switzerland involving 7006 participants reported that favorable SRH, healthy diet, less screen time, and good sleep quality were associated with optimistic health behaviors, with mental health and social support also providing critical roles [11].

Moreover, evidence suggests that well-being and SRH significantly influence the socio-demographic factors. In particular, SRH was strongly related to age, with increasing age correlated with poorer SRH [9,12]. A recent study conducted by Shalaby et al. (2024) indicated that those aged 40 years or younger, those with lower education, and those who are unmarried and unemployed were associated with low quality of life [9], while in a previous report of Parekh et al. (2018), poor well-being was significantly prevalent in female participants [13].

Examining the general population’s well-being and current health status is essential for understanding the overall health and quality of life, which could aid authorities in policy and program development, preventive strategies, health promotion, targeted intervention, and psychological evaluations. This evaluation could provide valuable insights into the current state and potential intervention areas for enhancing population well-being in Saudi Arabia. Hence, this study aims to assess the Saudi population’s well-being and self-rated health status and determine what factors influence these occurrences. We selected core socio-demographic factors, such as educational level, employment status, age, and gender, for this initial analysis because extensive literature demonstrates their strong influence on health outcomes and overall well-being [2]. These variables enable policymakers and public health practitioners to identify vulnerable subpopulations and tailor interventions accordingly. While this study focused on these fundamental demographic indicators, we acknowledge that other personal details—such as marital status and number of children—could provide a richer understanding of individual well-being. Future research will incorporate these additional variables to further clarify how personal and family contexts shape health perceptions and quality of life.

## 3. Study Objectives

To assess the overall level of well-being of the Saudi population using the WHO-05 index.

To identify the role of socio-demographic factors as one of the determinants of overall well-being.

To identify the association/relation [odds ratio] between socio-demographic and poor vs. good well-being.

### Study Hypothesis

Given the significant relationships identified in the literature between subjective well-being indices and socio-demographic factors, this study hypothesizes the following:

**H1:** 
*There will be a positive correlation between age and well-being, with older age groups demonstrating better well-being scores.*


**H2:** 
*Higher levels of educational attainment will positively correlate with better well-being scores as measured by the WHO-5 index.*


**H3:** 
*Gender and nationality will show significant associations with both WHO-5 and SRH, with males and non-Saudis demonstrating higher well-being and health ratings compared to females and Saudis.*


**H4:** 
*Being employed will be positively associated with self-rated health status (SRH).*


These hypotheses align with the study’s goal of identifying socio-demographic determinants significantly influencing well-being and self-rated health.

## 4. Study Design and Target Population

An analytical cross-sectional study was conducted among the general population in Saudi Arabia, including citizens and residents, all adult age groups (>18 years), and both genders.

## 5. Sampling Technique

A stratified random sampling was conducted to ensure homogeneity among each region (stratum) because of the heterogeneity of the entire population. This technique reduced the variation and resulted in a higher statistical precision. All 20 cities in Saudi Arabia were included. Proportional stratified random sampling was applied. Each region’s sample size is proportionate to that region’s population size to ensure equal representation of all regions. The following cities were included in our stratified sample, consisting of 100 samples each: Al Ahsa, Asir, Baha, Bisha, Eastern Province, Hafr Al-Batin, Hail, Jeddah, Jouf, Jazan, Madinah, Makkah, Najran, Northern, Border, Qunfuda, Qurayyat, Qassim, Riyadh, Tabuk, and Taif. Simple random sampling was used in each region to select samples. The selection of random samples was based on a computer-generated random numbers tool since the Mawid software app (https://www.moh.gov.sa/eServices/cards/Pages/Appointment-Booking-service.aspx, accessed on 1 August 2024) carried out this technique.

## 6. Sample Size

Using the Raosoft^®^ (http://www.raosoft.com/samplesize.html, accessed on 1 August 2024) sample size calculator, with a 95% confidence interval, a response distribution of 50%, and a 3% margin of error, the targeted sample size would be 1067 participants. Adjusting for the projected 10% attrition, the estimated final sample size for the general population survey is at least 1200 participants.

## 7. Measures

### 7.1. WHO-5 Well-Being Index

The Five-item Well-Being Index (WHO-5) stands as a validated instrument for the assessment of subjective psychological well-being over a specified period of two weeks. Originating in the 1990s, the validity of the WHO-5 has been confirmed through extensive studies across varied demographics and populations. The participants self-reported their well-being during the past two weeks. The scale has five items depicting feeling cheerful (Item 1: “I have felt cheerful and in good spirits”), feeling calm (Item 2: “I have felt calm and relaxed”), feeling active (Item 3: “I have felt active and vigorous”), feeling rested when waking up (Item 4: “I woke up feeling fresh and rested”), and feeling that one’s life is filled with exciting things (Item 5: “My daily life has been filled with things that interest me”) [14]. The response options ranged from 0 to 5, with 0 representing “at no time” and 5 “all the time”. In the present study, the WHO-5 Well-Being Index was calculated as the sum of the scores of the responses, ranging from 0 (the worst imaginable well-being) to 25 (the best imaginable well-being) [2]. The raw scores were transformed to a score from 0 to 100, with lower scores indicating worse well-being. A score of 50 or less suggests poor well-being [15]. This study used the Arabic version of the WHO-5, which is validated and tested [2]. Convergent and discriminant validity of WHO-5 yielded item measures between 0.591 and 0.710. All items are highly statistically significantly correlated with each other (*p* < 0.01). The average variance extracted (AVE) is 0.635, considering an AVE greater than 0.50; the results provide empirical evidence of convergent validity. The reliability test of the WHO-5 questionnaire has a Cronbach of 0.896 or 89.6%, indicating a very good internal consistency. Thus, the questionnaire was valid to use in this study.

### 7.2. Self-Rated Health (SRH)

Participants were asked to evaluate their own general health status using a standard self-rated health measure. The self-assessment involves a single item in which respondents describe their overall health from their personal perspective. Typical items ask individuals to rate their health generally along a 5-point scale, ranging from “excellent” to “poor”. For instance, the question might be formulated as follows: “In general, would you say your health is:” with the response options being 1: “Excellent”, 2: “Very good”, 3: “Good”, 4: “Fair”, and 5: “Poor”. Higher scores indicate poorer SRH. This subjective measure is widely recognized and utilized across various research domains due to its reliability and predictive validity for numerous health outcomes, such as mortality and chronic disease onset. The utilization of SRH in diverse populations and cultures has proven to be a valuable tool in assessing physical health, psychosocial, and well-being aspects, which are not easily measured by objective health indicators. Importantly, we utilized an Arabic version of the SRH measure, which has been validated and utilized in prior research in similar populations [16]. When conducting statistical tests, the “very good” and “good” categories were combined and named “healthy” groups, while “moderate”, “bad”, and “very bad” were combined and named “unhealthy” groups [17].

### 7.3. Socio-Demographic

Socio-demographic variables were selected for inclusion based on their established relationship with self-rated health status [17]. Characteristics included gender (male/female) and age (in years), nationality (Saudi/Non-Saudi), highest level of education (No formal education/Primary education/Elementary education/Secondary Education/University/higher education), occupational status (Governmental Employed/Private sector employed/Healthcare provider/governmental sector/Healthcare provider/private sector/Private Business or Freelancer/Retired/Student/Unemployed), geographical location (Place of residence (province) out of all 20 health cities in the kingdom). Then, all these geographical locations were categorized into 5 regions to ease analysis and comparability.

## 8. Data Collection

The data were collected through an online questionnaire using a Mom survey tool. The link to the survey was sent using SMS messages through the Seha app. The collection of data took place between February and June 2024. The study utilized the WHO-5 Well-Being Index, a validated tool assessing psychological well-being, and the self-rated health measure, where individuals rate their overall health perceptions. The questionnaire recorded basic demographic data and explored participants’ assessment regarding their well-being.

## 9. Statistical Analysis

The data were analyzed using the software program Statistical Packages for Software Sciences (SPSS) version 26 (Armonk, IBM Corporation, New York, NY, USA). Descriptive statistics were given as numbers and percentages (%) for all categorical variables, while continuous variables were calculated and summarized as mean and standard deviations. A Pearson correlation coefficient was used to determine the correlation between SRH and the well-being scores. Multiple logistic regression analyses were performed to ascertain the predictive factors of good well-being and healthy SRH with corresponding odds ratios as well as a 95% confidence interval. Values were considered significant with a *p*-value of less than 0.05.

## 10. Ethics and Confidentiality

All study participants were introduced to the study purpose and considered their participation after they started answering the questions. The survey forms were anonymous and did not include any identifiers or personal information of the participants to protect the confidentiality of the personal health information of participants. As the present study was an online survey-based report, participants were not required to provide written informed consent. The online survey’s cover page stated the study’s main objectives and informed the participants that their answers to the survey’s questions were used to assess the study’s objectives. Thus, participants who filled out the survey gave their consent to participate in the study. This study was approved by the Institutional Review Board (IRB), Riyadh Second Health Cluster, King Fahad Medical City, with IRB approval # 24-024E, and was approved on 11 January 2024.

## 11. Results

This national survey comprised 39,562 participants. Table 1 presents the socio-demographic characteristics of participants. A total of 30.6% were aged between 31 and 40 years old. Male respondents constitute almost two-thirds (65.1%) of participants. Most were Saudi nationals (83.8%) and university degree holders (56.1%). A total of 30.4% were private employees, and 33.4% lived in the Western Region. According to the city residency.

The majority of the respondents lived in Riyadh city (27.7%), followed by Eastern Province (17.7%) and Makkah (13.4%) (see Figure 1 below).

The perceived rating of current health according to SRH can be observed that 31.9% perceived their health status as very good, 45.5% indicated good, and only 0.9% indicated personal health as bad (see Figure 2 below).

Regarding the assessment of well-being using the WHO-5 Well-Being Index (Table 2), it can be observed that the highest rating was seen in the domain of feeling cheerful and in good spirits (mean score: 3.47), followed by feeling calm and relaxed (mean score: 3.21) and that daily life has been filled with interesting things (mean score 3.19). The total mean score of WHO-5 was 64.2 (SD 24). Accordingly, 26% were considered to have poor well-being, while the rest had good well-being (74%) (see Figure 3).

In the predictive model to identify the factors that influence good well-being (Table 3), adjusting for age, gender, nationality, and education, it was revealed that compared to the age group from 19 to 30 years, participants aged between 31 and 40 years had increased odds of having good well-being by at least 2.4 times higher (AOR = 2.409; 95% CI = 2.240–2.590; *p* < 0.001), participants aged between 41 and 50 years had increased odds of 1.68-fold higher (AOR = 1.682; 95% CI = 1.577–1.793; *p* < 0.001), and participants aged more than 50 years old had an increased odds of 1.29 times higher (AOR = 1.292; 95% CI = 1.208–1.382; *p* < 0.001). Also, respondents with better education were 1.43 times more likely to have a good well-being level than those with lower educational levels (AOR = 1.430; 95% CI = 1.362–1.502; *p* < 0.001). However, both the female gender (AOR = 0.794; 95% CI = 0.757–0.834; *p* < 0.001) and Saudis (AOR = 0.817; 95% CI = 0.764–0.874; *p* < 0.001) had a decreased chance of having good well-being by at least 21% and 18%, respectively, while respondents who were student/self-employed (AOR = 1.252; 95% CI = 1.130–1.388; *p* < 0.001) and non-healthcare providers (AOR = 1.451; 95% CI = 1.286–1.638; *p* < 0.001) were at increased odds of having good well-being by at least 1.25 and 1.45 times higher, respectively.

Multiple logistic regression analyses conducted in Table 4 to determine the influence of SRH in terms of participants’ socio-demographic characteristics indicate that, compared to participants with lower education, participants with higher education were 1.2 times more likely to have healthy SRH (AOR = 1.200; 95% CI = 1.140–1.263; *p* < 0.001). Also, compared to participants in the Central Region, participants who lived in the Western Region were at increased odds of having healthy SRH by at least 1.30 times higher (AOR = 1.302; 95% CI = 1.150–1.475; *p* < 0.001), increased odds of having healthy SRH by at least 1.3 times higher than those living in the Eastern Region (AOR = 1.303; 95% CI = 1.151–1.475; *p* < 0.001), increased by 1.44 times higher than those living in the Southern Region (AOR = 1.441; 95% CI = 1.266–1.640; *p* < 0.001), and 1.21 times higher those living in the Northern Region (AOR = 1.210; 95% CI = 1.056–1.387; *p* = 0.006). On the contrary, compared to male participants, female participants were at decreased odds of having healthy SRH by at least 28% (AOR = 0.679; 95% CI = 0.646–0.714; *p* < 0.001). Saudi participants also had a decreased chance of having healthy SRH by at least 12% compared to non-Saudi participants (AOR = 0.879; 95% CI = 0.821–0.941; *p* < 0.001). Similarly, compared to unemployed participants, participants who were healthcare providers were at decreased odds of having healthy SRH by almost 21% (AOR = 0.795; 95% CI = 0.728–0.868; *p* < 0.001) and decreased odds by at least 33% among those who were retired (AOR = 0.664; 95% CI = 0.578–0.763; *p* < 0.001). No significant effects were observed between SRH and age groups after adjustments to a regression model (*p* > 0.05).

## 12. Discussion

This study evaluated the level of well-being of Saudi residents by employing WHO-5 and SRH indices. To our knowledge, this is the first study in Saudi Arabia that assessed residents’ self-rated health status in a large population. Thus, the outcome of this study will be a great addition to the literature, given that mental well-being is a public health concern associated with various forms of physical and psychological disorders [18].

### 12.1. Well-Being Status

According to our results, more than one-fourth of our population was found to have poor well-being (mean WHO-5 score: 64.2 ± 24). These findings are consistent with the study of Caciula et al. (2019) [19]. The mean score for WHO-5 was 67.7, and low well-being constituted 25%. In contrast, a study by Shalaby et al. (2024) documented a WHO-5 mean score of 40.8 [10], lower than our report. The WHO-5 index has many advantages due to extensive previous research. It has been translated into different languages for more than 30 countries and has reported sufficient internal consistency in various research studies [2,20]. In addition, the results of this large cohort data would help authorities devise a program to enhance people’s mental states during vulnerable times.

### 12.2. Significant Predictor of Well-Being

Increasing age was associated with better well-being. In particular, the highest threshold was seen in the age group between 31 and 40 years (AOR = 2.13), followed by the age group between 41 and 50 years (AOR = 1.57) and the age group over 50 years (AOR = 1.13), validating the first hypothesis declared in our study. This contradicted the report of a study completed in Canada [9], wherein being 40 years or below was associated with low quality of life (QoL). Also, having a lower education, not being in a relationship, and being unemployed contributed to poor QoL. Supporting these reports, our study showed that lower education and unemployment could be associated with low well-being levels, but higher education, being a student, and employment were identified as independent significant predictors of good well-being, with an average increased threshold of at least 1.3-fold higher. This meets the criteria in our second hypothesis that higher education was positively correlated with better well-being.

In our third hypothesis, we predicted that gender and nationality have positive associations with WHO-5. This hypothesis was validated in our results as female participants and Saudi nationality had a decreased chance of having good well-being. Our study’s data suggest that females and Saudis were less associated with good well-being. There has been an indication that the association between these variables could be decreased by at least 16%. This indicates that the gender impacts may act differently based on the situational events. Consistent with these findings, a study completed in India [13] suggests that the poor well-being index was significantly higher in women, particularly among middle-aged women living in urban areas. Not opposing these results, Soldevila-Domenech et al. (2021) found that the critical risk factor for lower mental well-being was the lack of perceived social support, while health factors and self-perceived health were correlated highly to mental well-being status [21].

### 12.3. Self-Rated Health Status

Regarding the SRH index, 3.4% rated their health as bad or very bad, stratifying SRH into two groups: 77.4% were categorized as healthy and the rest were unhealthy (23.6%). This corroborated the study published in Canada [9] using a similar questionnaire; nearly half of the participants (45.9%) had an SRH rating of very good, followed by good (28.9%) and excellent (19.2%), and only 0.7% reported poor SRH. In contrast, a study conducted among Norwegian patients found that poor SRH was prevalent (48%), and this prevalence increased with an increasing number of symptoms. This suggests that the SRH of patients was proven to be lower than the general population and could be worse among patients suffering from multiple conditions [22].

### 12.4. Significant Predictor of SRH

This study suggests that respondents aged between 50 and 59 were at increased odds of reporting poor SRH by at least 84% times higher than the younger age group (age 35 to 49) and 119% for respondents aged between 60 and 69 years [9]. However, in an investigation completed by Iwata et al. (2023) [21], the mean SRH scores did not differ significantly between the age groups (<75 years vs. ≥75 years; *p* = 0.320). Among the age group 75 years or above, a more favorable economic level and better social engagement were associated with higher SRH scores [12]. In our study, however, after adjustments to multiple confounders, SRH showed no significant effect across the age groups (*p* > 0.05). This result necessitates further investigations. Study methodology, psychological adaptation, cultural influences, and the complex interplay of aging processes could play factors in the literature regarding differences in self-rated health across age groups.

Participants with better education were likelier to report being healthy than less educated participants, and the odds of reporting a favorable SRH could reach up to 115% (AOR = 1.151). Similarly, participants living outside the Central Region were also associated with better SRH than participants living in the Central Region. These reports are in agreement with the study of Fernandes et al. (2020) [23]. Poor health status was reported by women with lower education, lower family economic status, multiple chronic conditions, and more children, which corroborated the previous reports completed by Szwarcwald et al. (2005) [24].

On the contrary, we noticed that female and Saudi participants tended to report being unhealthy as compared to their male counterparts. Results from our adjusted model indicate 39% and 12% decreased chances of females and Saudis reporting they are healthy. This indicates that we validated the third hypothesis we set in this study, stating that male and non-Saudi participants were associated with better self-perceived health status. Likewise, respondents who were healthcare providers and retired could also display similar scenarios, suggesting a lower chance of reporting favorable outcomes, meeting the conditions of the fourth hypothesis we set in this study. This is consistent with the paper of Kjeldsberg et al. (2022), reporting a higher prevalence of poor SRH among women [22]. Notwithstanding these reports, Wu et al. (2013) found that the most significant factors of poor SRH include work pressure and life, poor spiritual conditions, and poor quality of interpersonal relationships [8].

### 12.5. Correlation Between Well-Being and Self-Rated Health Status

Moreover, we noted an existing correlation between WHO-5 and SRH scores (*p* < 0.001), suggesting that every increase in the score of SRH might correlate with every increase in the score of the WHO-5 index. In other words, the more favorable outcome of WHO-5 could be associated with the more favorable outcome of SRH. This agrees with the study published in Switzerland [11]. Positive health behaviors were linked to consistently positive self-rated health, less screen time, a healthy diet, and good sleep quality. Social support and mental health also had significant roles. However, in Finland [7], during the 10-year follow-up, poor SRH and objective health (OH) predicted about four times higher risk for mortality than those with good health. Accounting for these scenarios, in a 27-year follow-up, OH was a greater mortality predictor than SRH. The author further suggested that periodic SRH collection could boost focus on patient-centered care. These variations could be attributed mainly to the type of tool used, the population’s characteristics, and theoretical frameworks.

## 13. Theoretical and Practical Contribution

The theoretical and practical implications of the constructs are significant across various domains, such as public health, psychology, sociology, and policymaking. Understanding the association between well-being and SRH and the influence of demographic factors is critical in formulating intervention programs. The findings of this study could also provide practical implications for government agencies in terms of health monitoring, early intervention, policy development, education, and public awareness. Addressing the gaps across each domain and applying a holistic approach may improve the overall health of the general population.

## 14. Study Strength

This study provides evidence that higher well-being correlates with improved self-rated health status. This large cohort study also offers strong evidence of an association between WHO-5 and SRH in terms of key demographic variables such as age, gender, and occupation. However, integrating objective health data and addressing the subjective nature of the constructs are vital to improving the strength of this research.

## 15. Study Limitation

The results of this study account for several limitations. First, confirmatory factor analysis (CFA) was not performed in this study, and we assumed that WHO-5 has only five items that may not necessarily be needed to perform CFA, and we followed the criteria given by WHO. Second, a large sample may have strong statistical power and reliability but oftentimes encounters data management challenges, the risk of overgeneralization, and statistical misrepresentation. Third, key demographic factors, such as marital status and number of children, were not measured in this study, which should be considered in future research. Lastly, a cross-sectional survey could be prone to bias, is unable to determine cause and effect, and cannot be used to measure behavior over time; hence, prospective studies could provide better insights into the well-being and self-rated status of the general population.

## 16. Conclusions

Both WHO-5 and SRH outcomes led to satisfactory well-being for this national survey in Saudi Arabia. Increasing age, better education, being an employee, and being a student may result in a favorable health status among this population group. However, more focus is needed on Saudi women who are either healthcare providers or retired, as they may exhibit low quality of life. This study also provides evidence of strong agreement between WHO-5 and SRH. Hence, more investigations are warranted to confirm this existing association. We emphasize the importance of healthy lifestyle preferences and social support to promote favorable well-being. The findings of this study could be used for future reference by our authorities to develop targeted interventions and strategies to improve the well-being of the increasing population in Saudi Arabia.

## Figures and Tables

**Figure 1 healthcare-13-00310-f001:**
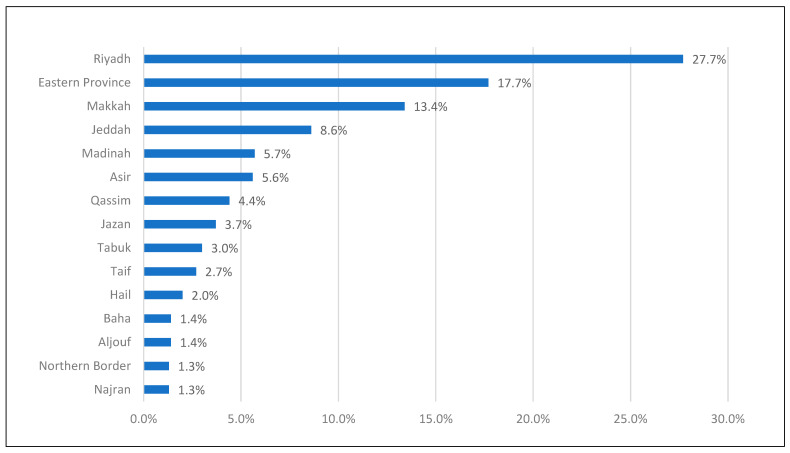
Distribution of responses according to the city of residence.

**Figure 2 healthcare-13-00310-f002:**
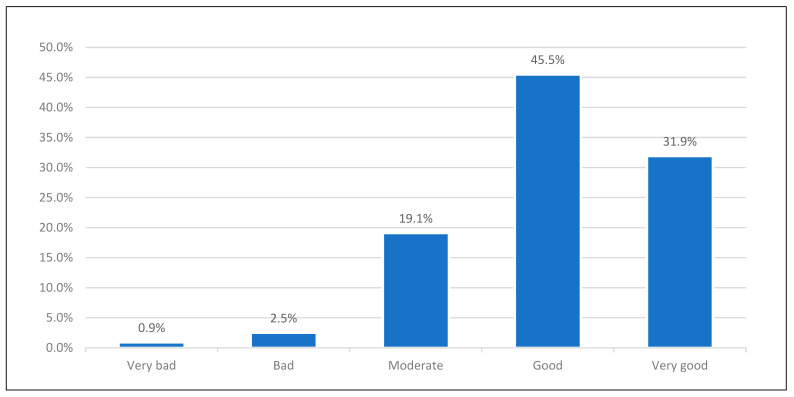
Perceived rating of current health based on SRH.

**Figure 3 healthcare-13-00310-f003:**
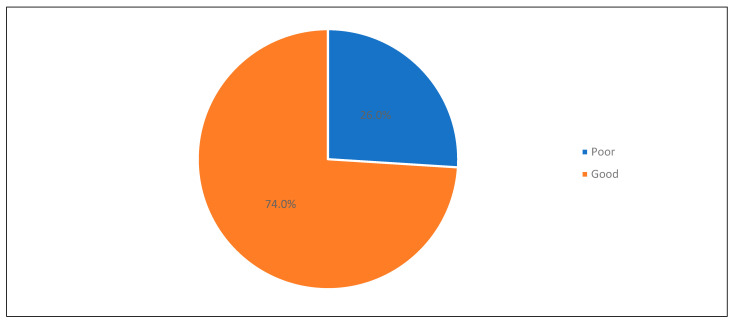
Level of well-being.

**Table 1 healthcare-13-00310-t001:** Socio-demographic characteristics of participants^(n=39,562)^.

Study Variables	N (%)
Age group	
19–30 years	6516 (16.5%)
31–40 years	12,102 (30.6%)
41–50 years	10,510 (26.6%)
>50 years	10,434 (26.4%)
Gender	
Male	25,743 (65.1%)
Female	13,819 (34.9%)
Nationality	
Non-Saudi	6390 (16.2%)
Saudi	33,172 (83.8%)
Highest level of education	
Uneducated	109 (0.30%)
Primary School	641 (01.6%)
Elementary School	1624 (04.1%)
Secondary School	9810 (24.8%)
University degree	22,178 (56.1%)
Higher Education	5200 (13.1%)
Occupational status	
Unemployed	7750 (19.6%)
Student	1323 (03.3%)
Government employee	12,024 (30.4%)
Private employee	8586 (21.7%)
Government healthcare provider	1479 (03.7%)
Private healthcare provider	798 (02.0%)
Self-employed	1273 (03.2%)
Retired	6329 (16.0%)
Region of residence	
Central Region	12,706 (32.1%)
Western Region	13,223 (33.4%)
Eastern Region	6989 (17.7%)
Southern Region	4754 (12.0%)
Northern Region	1890 (04.8%)

**Table 2 healthcare-13-00310-t002:** Assessment of participants’ well-being using the WHO-5 Well-Being Index^(n=39,562)^.

Domain	Mean ± SD
I have felt cheerful and in good spirits	3.47 ± 1.37
I have felt calm and relaxed	3.21 ± 1.41
I have felt active and vigorous	3.11 ± 1.39
I woke up feeling fresh and rested	3.06 ± 1.49
My daily life has been filled with things that interest me	3.19 ± 1.48
Total composite WHO-5 score *	64.2 ± 24.0
Level of well-being	
Poor (score < 50)	10,292 (26.0%)
Good (score ≥ 50)	29,270 (74.0%)

Response has a range from “At no time”, coded with 0, to “All of the time”, coded with 5. * To obtain a percentage score ranging from 0 to 100, the raw score is multiplied by 4.

**Table 3 healthcare-13-00310-t003:** Multiple logistic regression analysis to identify the factors that influence good well-being^(n=39,562)^.

Factor	Level of Well-Being		AOR (95% CI)	*p*-Value
	Good N (%)^(n=29,270)^	Poor N (%)^(n=10,292)^		
Age group				
19–30 years	4091 (14.0%)	2425 (23.6%)	Ref	
31–40 years	8674 (29.6%)	3428 (33.3%)	2.409 (2.240–2.590)	<0.001 **
41–50 years	8048 (27.5%)	2462 (23.9%)	1.682 (1.577–1.793)	<0.001 **
>50 years	8457 (28.9%)	1977 (19.2%)	1.292 (1.208–1.382)	<0.001 **
Gender				
Male	19,682 (67.2%)	6061 (58.9%)	Ref	
Female	9588 (32.8%)	4231 (41.1%)	0.794 (0.757–0.834)	<0.001 **
Nationality				
Non-Saudi	5059 (17.3%)	1331 (12.9%)	Ref	
Saudi	24,211 (82.7%)	8961 (87.1%)	0.817 (0.764–0.874)	<0.001 **
Highest level of education				
Secondary or below	8530 (29.1%)	3654 (35.5%)	Ref	
University or higher	20,740 (70.9%)	6638 (64.5%)	1.430 (1.362–1.502)	<0.001 **
Occupational status				
Unemployed	5166 (17.6%)	2584 (25.1%)	Ref	
Student/Self-employed	1651 (05.6%)	945 (09.2%)	1.252 (1.130–1.388)	<0.001 **
Non-healthcare provider	15,516 (53.0%)	5094 (49.5%)	1.451 (1.286–1.638)	<0.001 **
Healthcare provider	1817 (06.2%)	460 (04.5%)	1.087 (0.993–1.190)	0.070
Retired	5120 (17.5%)	1209 (11.7%)	0.899 (0.783–1.031)	0.127
Region of residence				
Central Region	9441 (32.3%)	3265 (31.7%)	Ref	
Western Region	9822 (33.6%)	3401 (33.0%)	1.008 (0.901–1.127)	0.892
Eastern Region	5209 (17.8%)	1780 (17.3%)	1.010 (0.904–1.129)	0.856
Southern Region	3405 (11.6%)	1349 (13.1%)	0.986 (0.877–1.109)	0.820
Northern Region	1393 (04.8%)	497 (04.8%)	1.096 (0.970–1.238)	0.143

The odds ratios were calculated and referenced by the “good well-being” group and adjusted with age, gender, nationality, and education. AOR—Adjusted Odds Ratio; CI—confidence interval. ** Significant at *p* < 0.05 level.

**Table 4 healthcare-13-00310-t004:** Multiple logistic regression analysis to identify the factors that influence the healthy self-rated health status of participants^(n=39,562)^.

Factor	Self-Rated Health Status		AOR (95% CI)	*p*-Value
	Healthy N (%)^(n=30,629)^	Unhealthy N (%)^(n=8933)^		
Age group				
19–30 years	4900 (16.0%)	1616 (18.1%)	Ref	
31–40 years	9411 (30.7%)	2691 (30.1%)	1.049 (0.972–1.131)	0.217
41–50 years	8150 (26.6%)	2360 (26.4%)	0.974 (0.913–1.039)	0.424
>50 years	8168 (26.7%)	2266 (25.4%)	1.007 (0.943–1.076)	0.829
Gender				
Male	20,557 (67.1%)	5186 (58.1%)	Ref	
Female	10,072 (32.9%)	3747 (41.9%)	0.679 (0.646–0.714)	<0.001 **
Nationality				
Non-Saudi	5144 (16.8%)	1246 (13.9%)	Ref	
Saudi	25,485 (83.2%)	7687 (86.1%)	0.879 (0.821–0.941)	<0.001 **
Highest level of education				
Secondary or below	9194 (30.0%)	2990 (33.5%)	Ref	
University or higher	21,435 (70.0%)	5943 (66.5%)	1.200 (1.140–1.263)	<0.001 **
Occupational status				
Unemployed	5618 (18.3%)	2132 (23.9%)	Ref	
Student/Self-employed	1900 (06.2%)	696 (07.8%)	0.929 (0.839–1.028)	0.154
Non-healthcare provider	16,399 (53.5%)	4211 (47.1%)	1.014 (0.896–1.147)	0.827
Healthcare provider	1866 (06.1%)	411 (04.6%)	0.795 (0.728–0.868)	<0.001 **
Retired	4846 (15.8%)	1483 (16.6%)	0.664 (0.578–0.763)	<0.001 **
Region of residence				
Central Region	9827 (32.1%)	2879 (32.2%)	Ref	
Western Region	10,231 (33.4%)	2992 (33.5%)	1.302 (1.150–1.475)	<0.001 **
Eastern Region	5287 (17.3%)	1702 (19.1%)	1.303 (1.151–1.475)	<0.001 **
Southern Region	3740 (12.2%)	1014 (11.4%)	1.441 (1.266–1.640)	<0.001 **
Northern Region	1544 (05.0%)	346 (03.9%)	1.210 (1.056–1.387)	0.006 **

The odds ratios were calculated and referenced by the “healthy” group and adjusted with age, gender, nationality, and education. AOR—Adjusted Odds Ratio; CI—confidence interval. ** Significant at *p* < 0.05 level.

## Data Availability

The original contributions presented in this study are included in the article. Further inquiries can be directed to the corresponding author.

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
