# Peer review of "Assessing Population Well-Being in Saudi Arabia: A Comprehensive National Survey Using the WHO-05 Index and Self-Rated Health Metrics"

_healthcare, 2025, doi:10.3390/healthcare13030310_

Round 1

Reviewer 1 Report

Comments and Suggestions for Authors

1.In the abstract, in the Result section, concerning the participants’ descriptive statistics, “In total, 39562 participants responded to our survey (male 65.1% vs female 34.9%). 30.6% 27 were between 31 and 40 years old, with most of the population being Saudi nationals (83.8%).” would be better put in the Method section.

2.In the abstract, lines 31-32, “Independent predictors for good well-being include … being unemployed.” However, in the line 35, “employment, … were the protective factors of good well-being.” These two sentences were conflicting. The authors should recheck the analysis outcomes and related writing.

3.Figure 3 was not necessary. Delete it would be better.

4.It was better to use the sample to present the information of reliability and validity of WHO-5 and SRH.

5.Under Table 3, the denotation ‘The odds ratios were calculated referenced by the "good well-being” group’ would be misunderstanding. Did the author want to express that the odds was calculated on basis of "good well-being” relative to “poor well-being” ? Besides, It was better to express that the reference group for categorical or ordinal predictors as “Ref.”.

6.The title of Table 3 and 4 “Univariate and multivariate analyses” was misunderstanding. For only one dependent variable, using “univariate or not need to report univariate” Besides, for at least two predictors, using “multiple,” such as “multiple logistic regression.” Furthermore, please denote under the Tables which variables were controlled for AOR. Finally, please to denote these factors “age group, gender, nationality,…Region of residence” were simultaneously put into the logistic regression or separately (with which controlled variables?).

7. Under Table 4, please denote how to separate the two groups “healthy” and “unhealthy”

Author Response

Reviewer comments:
1. In the abstract, in the Result section, concerning the participants’ descriptive statistics, “In total, 39562 participants responded to our survey (male 65.1% vs female 34.9%). 30.6% 27 were between 31 and 40 years old, with most of the population being Saudi nationals (83.8%).” would be better put in the Method section.

2.In the abstract, lines 31-32, “Independent predictors for good well-being include … being unemployed.” However, in the line 35, “employment, … were the protective factors of good well-being.” These two sentences were conflicting. The authors should recheck the analysis outcomes and related writing.

3.Figure 3 was not necessary. Delete it would be better.

4.It was better to use the sample to present the information of reliability and validity of WHO-5 and SRH.

5.Under Table 3, the denotation ‘The odds ratios were calculated referenced by the "good well-being” group’ would be misunderstanding. Did the author want to express that the odds was calculated on basis of "good well-being” relative to “poor well-being” ? Besides, It was better to express that the reference group for categorical or ordinal predictors as “Ref.”.

6.The title of Table 3 and 4 “Univariate and multivariate analyses” was misunderstanding. For only one dependent variable, using “univariate or not need to report univariate” Besides, for at least two predictors, using “multiple,” such as “multiple logistic regression.” Furthermore, please denote under the Tables which variables were controlled for AOR. Finally, please to denote these factors “age group, gender, nationality,…Region of residence” were simultaneously put into the logistic regression or separately (with which controlled variables?).

7. Under Table 4, please denote how to separate the two groups, “healthy” and “unhealthy”

Authors Comments: 

Comments 1 and 2:

The abstract has been revised

Comment 3:

Figure 3 was deleted as per the comments

Comment 4:

Validity and reliability have been added as highlighted for WHO-5. However, SRH was not measurable using a reliability test.

Comments 5 and 6:

The titles of Tables 3 and 4 have been revised. The enter method was used to adjust for multiple confounders, where all variables were added to the model. This information is added in the footnote of each table.

Comment 7:

The criteria for self-rated health status are presented in the methodology-measures section.

Reviewer 2 Report

Comments and Suggestions for Authors

Dear authors,

Thank you so much for your work. The manuscript presents an interesting topic. However, the authors are required to address several issues before the article is deemed suitable for publication. I wish the authors well in their efforts and offer the following concerns for their consideration.

1-     The introduction section should be thought of as the “visiting card” of the study, making the purpose and contribution of the investigation clear. However, in its current state, the introduction does not serve its purpose given that it is not entirely clear what this study adds to the literature, why it is essential, what is already known in general terms, and how authors intend to contribute. In particular, the authors should briefly introduce the relationship between the variables (well-being and self-rated health metrics). The authors should clarify the literature gap in the introduction, explain why they have chosen these variables (provide a clear description of the problem), state the study's main aim, and state how the authors intend to contribute.

2-     I suggest that the authors remove the information related to scales in the introduction section, especially: “This measure provides participants with an opportunity to evaluate their own health from a personal standpoint. Participants  typically respond to a straightforward question, such as "In general, how would you rate your health?", using a 5-point scale with options ranging from 'excellent' to 'poor' (7).”

3-     The objectives defined should be properly justified. For example, why did the authors decide to establish a relationship between socio-demographic variables and well-being? On the other hand, given that this is a study of the well-being of adults in general, why was other personal information about the participants not considered? For example, the participant's marital status and number of children.

4-     The use of a subheading about outcome utilization is very confusing. I suggest the authors consider removing this subheading.

5-      The relationship between the variables should be presented in a specific section called a literature review or hypothesis development.

6-     Since this is a quantitative study, it is not clear why the authors chose not to define hypotheses. This option should be duly justified.

7-     Authors should provide information about the period of data collection and how participants were contacted.

8-     The authors should provide information about the Cronbach alphas of the scales.

9-     Authors must provide the results of scale validation (convergent and discriminant analyses), as well as descriptive statistics of the scales and correlation analysis between variables. The authors present descriptive statistics of the items, not of the scales under study (well-being and self-rated health metrics).

10-  The authors did not perform confirmatory factor analysis, this issue must be justified or presented as a limitation of the study.

11-  The authors did not mention the procedures used to minimize the common method bias. This is a critical issue that should be addressed.

12-  Authors should present a more robust discussion section that reflects the relationship between well-being and self-rated health metrics.

13-  The authors should add and substantiate the theoretical and practical contributions of the study. On the other hand, authors should provide a clear description of the limitations of the study and how they can be addressed in future studies.

Author Response

Reviewers comments: 

1-     The introduction section should be thought of as the "visiting card" of the study, making the purpose and contribution of the investigation clear. However, in its current state, the introduction does not serve its purpose given that it is not entirely clear what this study adds to the literature, why it is essential, what is already known in general terms, and how authors intend to contribute. In particular, the authors should briefly introduce the relationship between the variables (well-being and self-rated health metrics). The authors should clarify the literature gap in the introduction, explain why they have chosen these variables (provide a clear description of the problem), state the study's main aim, and state how the authors intend to contribute.

2-     I suggest that the authors remove the information related to scales in the introduction section, especially: "This measure provides participants with an opportunity to evaluate their own health from a personal standpoint. Participants  typically respond to a straightforward question, such as "In general, how would you rate your health?", using a 5-point scale with options ranging from 'excellent' to 'poor' (7)."

3-     The objectives defined should be properly justified. For example, why did the authors decide to establish a relationship between socio-demographic variables and well-being? On the other hand, given that this is a study of the well-being of adults in general, why was other personal information about the participants not considered? For example, the participant's marital status and number of children.

4-     The use of a subheading about outcome utilization is very confusing. I suggest the authors consider removing this subheading.

5-      The relationship between the variables should be presented in a specific section called a literature review or hypothesis development.

6-     Since this is a quantitative study, it is not clear why the authors chose not to define hypotheses. This option should be duly justified.

7-     Authors should provide information about the period of data collection and how participants were contacted.

8-     The authors should provide information about the Cronbach alphas of the scales.

9-     Authors must provide the results of scale validation (convergent and discriminant analyses), as well as descriptive statistics of the scales and correlation analysis between variables. The authors present descriptive statistics of the items, not of the scales under study (well-being and self-rated health metrics).

10-  The authors did not perform confirmatory factor analysis, this issue must be justified or presented as a limitation of the study.

11-  The authors did not mention the procedures used to minimize the common method bias. This is a critical issue that should be addressed.

12-  Authors should present a more robust discussion section that reflects the relationship between well-being and self-rated health metrics.

13-  The authors should add and substantiate the theoretical and practical contributions of the study. On the other hand, authors should provide a clear description of the limitations of the study and how they can be addressed in future studies.

Authors comments:

Comments 1:

The introduction has been revised

Comment 2:

The texts, as suggested, have been removed from the introduction

Comment 3:

The study's objective has been fixed. The missing demographic variables, which were not included in this study, add to the limitations.

Comment 4:

The utilization subsection has been removed.

Comment 5:

The hypothesis has been added

Comment 6:

The hypothesis has been added.

Comment 7:

dates of data collection have been added 

Comment 8:

Cronbach alpha has been added to the texts as highlighted.

Comment 9:

Discriminant analysis and item correlation matrix have been added to the WHO-5 questionnaire under methodology (measure subsection) as highlighted.

Comment 10:

Added to the limitation of the study regarding comments about confirmatory factor analysis.

Comment 11:

Added method to adjust for multiple confounders in the footnote of tables 3 and 4 (regression models).

Comment 12:

The correlation between well-being and SRH is clearly discussed in the discussion section. I added some contexts, which are highlighted in yellow.

Comments 13:
study limitations have been addedd

Reviewer 3 Report

Comments and Suggestions for Authors Abstract: -when have you carried out study and how? Introduction why have you chosen Well-Being Index and Self-Rated Health (SRH) scales? They are applied to mainly healthcare.Here, you applied general population. Are there any other scales for these purposes? Give some information about Region of residences? Economic powers? Education? Health indicators etc. Measures -Has anyone carried out reliability and validity study of WHO-5 Well-Being Index and Self-Rated Health (SRH) studies in Saudi Arabia in previous studies? have they translated to in Arabic? Data collection:When have you carried out that study? Hoe have you prepared online questionnaire? Google form or? How have you distributed questionary and how many to each city and region?Give some info about cities and regions. Ethics and Confidentiality: did you get any ethical permission from any university? Sociodemographic characteristics of participants: add name of cities and survey numbers Table 4: Univariate and multivariate analyses to identify the factors that influence self-rated health status of participants: I would suggest you to add cities too. Why have you chosen Univariate and multivariate analyses? T-test or ANOVA may be more suitable.You have a large sample size. Line 281: highest threshold?Values? Line 287-289: Reasons of differences? Line 316:84% times higher than the younger age? times higher? You may improve your Correlation by adding age, education level Limitations of your study? Your contribution?Future studies? You may improve literature by adding some more recent studies    

Author Response

Reviewer comments:
 Abstract: -when have you carried out study and how? Introduction why have you chosen Well-Being Index and Self-Rated Health (SRH) scales? They are applied to mainly healthcare.Here, you applied general population. Are there any other scales for these purposes? Give some information about Region of residences? Economic powers? Education? Health indicators etc. Measures -Has anyone carried out reliability and validity study of WHO-5 Well-Being Index and Self-Rated Health (SRH) studies in Saudi Arabia in previous studies? have they translated to in Arabic? Data collection:When have you carried out that study? Hoe have you prepared online questionnaire? Google form or? How have you distributed questionary and how many to each city and region?Give some info about cities and regions. Ethics and Confidentiality: did you get any ethical permission from any university? Sociodemographic characteristics of participants: add name of cities and survey numbers Table 4: Univariate and multivariate analyses to identify the factors that influence self-rated health status of participants: I would suggest you to add cities too. Why have you chosen Univariate and multivariate analyses? T-test or ANOVA may be more suitable.You have a large sample size. Line 281: highest threshold?Values? Line 287-289: Reasons of differences? Line 316:84% times higher than the younger age? times higher? You may improve your Correlation by adding age, education level Limitations of your study? Your contribution?Future studies? You may improve literature by adding some more recent studies    

Authors comments:

We have chosen categorical outcome variables for WHO-5 and SRH to identify the independent significant predictors of the target groups. T-tests and One-way ANOVA may not provide such outcomes. Logistics regression also provides the odds ratio, while the t-test and ANOVA test do not. Also, logistics regression can be adjusted for multiple confounders. I added more context about the differences in SRH across the age groups in the literature. The Arabic translation was clearly mentioned in the methodology with reference 2. The questionnaire was distributed using SMS messages with a link shared through Seha Apps (MoH app for patients to book their visits..etc) across Saudi Arabia. Also added the distribution of cities as suggested by one of the reviewers. Added literature reviews section. Other comments has been already addressed in other reviewers comments and can be found in the reviewed manuscript.

Round 2

Reviewer 1 Report

Comments and Suggestions for Authors

1. Two "Table 1" of the Table title were found. Please also check the contents.

2."logistic regression" would be better than "logistics regression"

Author Response

Reviewer comments: 

1. Two "Table 1" of the Table title were found. Please also check the contents.

2."logistic regression" would be better than "logistics regression"

Response comments: 

This has been revised as highlighted; the first table in the method was removed and written as text, an
 Logistic regression has been rectified as suggested.

Reviewer 2 Report

Comments and Suggestions for Authors

Dear authors,

I congratulate the authors for their efforts to improve the manuscript. However, I still have doubts regarding the clarification of the following aspects:

1-      The reformulation of the objective of the study leaves the idea of the relationship between working conditions, service quality and performance intact. However, the keywords do not mention service quality and performance, which is an issue that requires reformulation.

2-      The introduction section of the manuscript requires a redesign for the following reasons:

a). The gap in the literature is not clearly defined, nor is it evident how the authors intend to address it.

b). The introduction does not present a clear definition of the variables under study. It is not evident how the authors define the terms "WHO-5 Index," "self-rated health status," and "well-being ".

c) The objective of the study was not presented or justified by the authors. This is a crucial issue that requires resolution.

3- The preceding issue, as previously identified in the initial review, remains unresolved: “The objectives defined should be properly justified. For example, why did the authors decide to establish a relationship between socio-demographic variables and well-being? On the other hand, given that this is a study of the well-being of adults in general, why was other personal information about the participants not considered? For example, the participant's marital status and number of children”.

4-     The authors employed an inappropriate definition of hypotheses in the context of correlational studies. To provide further clarification and justification of the definition of hypotheses, the following examples are provided:

a)      Rego, A., Yam, K. C., Owens, B. P., Story, J. S. P., Pina e Cunha, M., Bluhm, D., & Lopes, M. P. (2017). Conveyed Leader PsyCap Predicting Leader Effectiveness Through Positive Energizing. Journal of Management, 014920631773351. doi:10.1177/0149206317733510 

b)     Esteves, T., & Lopes, M. P. (2016). Crafting a Calling. Journal of Career Development, 44(1), 34–48. doi:10.1177/0894845316633789

5-     Authors must provide the results of scale validation (convergent and discriminant analyses), as well as descriptive statistics of the scales and correlation analysis between variables. The authors present descriptive statistics of the items, not of the scales under study (well-being and self-rated health metrics).

6-     The authors did not mention the procedures used to minimize the common method bias. This is a critical issue that should be addressed. The authors added the following information: "The enter method was used to adjust for multiple confounders in the regression model". However, it does not present a discussion of common method bias.

7-     The discussion of the manuscript should reflect the main questions raised in the study. If the authors define hypotheses, they should add a discussion of whether or not the hypotheses were confirmed and provide appropriate justification. Since this is a study conducted in Saudi Arabia, the authors should present a discussion of the issues related to the generalizability of the study results.

8-     The authors should add and substantiate the theoretical and practical contributions of the study. On the other hand, authors should provide a clear description of the limitations of the study and how they can be addressed in future studies.

9-     The subtitle called Recommendation does not reflect the appropriate standard for scientific inquiry. Therefore, I suggest that the authors consider removing it.

Author Response

Comments

  1. The reformulation of the objective of the study leaves the idea of the relationship between working conditions, service quality and performance intact. However, the keywords do not mention service quality and performance, which is an issue that requires reformulation.

Response

I added the objectives of the study

Comments

  1. The introduction section of the manuscript requires a redesign for the following reasons:
  2. The gap in the literature is not clearly defined, nor is it evident how the authors intend to address it.
  3. The introduction does not present a clear definition of the variables under study. It is not evident how the authors define the terms "WHO-5 Index," "self-rated health status," and "well-being ".
  4. The objective of the study was not presented or justified by the authors. This is a crucial issue that requires resolution.

Response

I added some contexts in the introduction (highlighted. I added the objectives of the study. 

Comments

  1. The preceding issue, as previously identified in the initial review, remains unresolved: "The objectives defined should be properly justified. For example, why did the authors decide to establish a relationship between socio-demographic variables and well-being? On the other hand, given that this is a study of the well-being of adults in general, why was other personal information about the participants not considered? For example, the participant's marital status and number of children".

Response

The missing key variables, such as marital status and number of children, were mentioned in the limitations of the study.

Comments

  1. The authors employed an inappropriate definition of hypotheses in the context of correlational studies. To provide further clarification and justification of the definition of hypotheses, the following examples are provided:
  2. Rego, A., Yam, K. C., Owens, B. P., Story, J. S. P., Pina e Cunha, M., Bluhm, D., & Lopes, M. P. (2017). Conveyed Leader PsyCap Predicting Leader Effectiveness Through Positive Energizing. Journal of Management, 014920631773351. doi:10.1177/0149206317733510
  3. Esteves, T., & Lopes, M. P. (2016). Crafting a Calling. Journal of Career Development, 44(1), 34–48. doi:10.1177/0894845316633789

Comments

  1. Authors must provide the results of scale validation (convergent and discriminant analyses), as well as descriptive statistics of the scales and correlation analysis between variables. The authors present descriptive statistics of the items, not of the scales under study (well-being and self-rated health metrics).

Response

The results of the convergent and discriminant validity of WHO-5 have been added in the methodology-measures section.

Comments

  1. The authors did not mention the procedures used to minimize the common method bias. This is a critical issue that should be addressed. The authors added the following information: "The enter method was used to adjust for multiple confounders in the regression model". However, it does not present a discussion of common method bias.

Response

The multiple logistic regression analysis method has been reclassified to adjust for multiple confounders such as age, gender, nationality, and education.

Comments

  1. The discussion of the manuscript should reflect the main questions raised in the study. If the authors define hypotheses, they should add a discussion of whether or not the hypotheses were confirmed and provide appropriate justification. Since this is a study conducted in Saudi Arabia, the authors should present a discussion of the issues related to the generalizability of the study results.

Response                         

Comments

  1. The authors should add and substantiate the theoretical and practical contributions of the study. On the other hand, authors should provide a clear description of the limitations of the study and how they can be addressed in future studies.

Response

Added the theoretical and practical contribution section and added some context to the limitations of the study.

Comments

  1. The subtitle called Recommendation does not reflect the appropriate standard for scientific inquiry. Therefore, I suggest that the authors consider removing it.

Response

This subsection has been deleted as suggested.

Reviewer 3 Report

Comments and Suggestions for Authors Abstract: -Add abstract when have you carried out study and how? -Line 88-99: Give some explanation for your Hypothesises from past studies. -No need table 1, you may verbally stated that min. 100 sample required for each city, totally 2000 and write names of all cities.    

Author Response

Reviewer comments:
-Add abstract when have you carried out study and how? -Line 88-99: Give some explanation for your Hypothesises from past studies. -No need table 1, you may verbally stated that min. 100 sample required for each city, totally 2000 and write names of all cities.    

Response comments: 
The revised submitted manuscript has been reviewed and updated according to the comments mentioned.